# Evaluation of a Norcantharidin Nanoemulsion Efficacy for Treating B16F1-Induced Melanoma in a Syngeneic Murine Model

**DOI:** 10.3390/ijms26031215

**Published:** 2025-01-30

**Authors:** Gabriel Martínez-Razo, Patrícia C. Pires, Angélica Avilez-Colin, María Lilia Domínguez-López, Francisco Veiga, Eliezer Conde-Vázquez, Ana Cláudia Paiva-Santos, Armando Vega-López

**Affiliations:** 1Laboratorio de Toxicología Ambiental, Escuela Nacional de Ciencias Biológicas, Instituto Politécnico Nacional, Unidad Profesional Zacatenco, Mexico City 07738, Mexico; 2Department of Pharmaceutical Technology, Faculty of Pharmacy of the University of Coimbra, University of Coimbra, Azinhaga Sta. Comba, 3000-548 Coimbra, Portugal; patriciapires@ff.uc.pt (P.C.P.);; 3LAQV, REQUIMTE, Department of Pharmaceutical Technology, Faculty of Pharmacy of the University of Coimbra, University of Coimbra, Azinhaga Sta. Comba, 3000-548 Coimbra, Portugal; 4Health Sciences Research Centre (CICS-UBI), University of Beira Interior, Av. Infante D. Henrique, 6200-506 Covilhã, Portugal; 5Hospital Bicentenario de la Independencia del Instituto de Salud de Trabajadores del Estado ISSSTE, Ciruelos 4, Lázaro Cárdenas, Tultitlán de Mariano Escobedo 54916, Mexico

**Keywords:** melanoma, nanoemulsion, nanosystems, nanotechnology, norcantharidin, topical administration

## Abstract

Melanoma, a lethal type of cancer originating from melanocytes, is the leading cause of death among skin cancers. While surgical excision of the lesions is the primary treatment for melanoma, not all cases are candidates for surgical procedures. New treatments and complementary options are necessary, given the increasing diagnosis rate. In the present study, a norcantharidin-containing nanoemulsion was developed and evaluated in vivo using a syngeneic graft murine model. Norcantharidin is the demethylated analog of cantharidin, known for its anticancer properties. Our model contemplates surgical excision surgery simulating the standard treatment and the role of the nanoemulsion as a potential adjuvant therapy. We observed a significant decrease in the growth rate of the melanoma lesion in the treated groups compared to the control group, both at the 20th and 30th days of treatment. Moreover, we evaluated the drug bioavailability in serum samples, and the results showed that norcantharidin was detectable in a range of 0.1 to 0.18 mg/mL in the treated groups. Furthermore, histopathological analysis was performed on the amputated tumors, where significant differences were found regarding size, mitosis rate, lymphocytic infiltration, and multispectral quantitative image analysis compared to the control group. If more clinical studies are conducted, the norcantharidin-containing nanoemulsion could be a potential alternative or adjuvant therapy. Topical nanosystems can become or complement standard therapies, which is needed as melanoma affects not only in terms of mortality but also the patient’s morbidity and life quality.

## 1. Introduction

Melanoma is a malignant neoplasm caused by the uncontrolled growth of the skin’s pigment-producing cells, melanocytes [1]. According to the International Agency for Research on Cancer’s GLOBOCAN, in 2018, melanoma caused 60,712 deaths, and it is estimated that 466,914 new cases will be diagnosed in 2040 [2]. Melanoma-related mortality is relatively stable with a tendency to increase, but much slower than the incidence [3]. This discrepancy may be due to overdiagnosis, which correlates with an increase in early-stage diagnoses [4]. Furthermore, melanoma-related deaths commonly arise from rapidly progressing melanomas that are rarely detected during screening processes [5]. A favorable prognosis is attained only through early diagnosis, followed by the prompt excision of cutaneous lesions. The clinical ABCD rule is based on four clinical morphologies of melanoma: (1) Asymmetry, (2) Border irregularity, (3) Color variation, and (4) Diameter greater than 6 mm [6]; it has been established as a framework for distinguishing melanomas from benign pigmented skin lesions, forming the basis of current clinical diagnosis.

Melanoma lesions with subtle signs are sometimes mistaken for benign lesions, and conversely, benign moles are sometimes misdiagnosed as melanomas, leading to unnecessary biopsies [7]. Nonetheless, surgical excision offers a convenient therapeutic approach and, when viable, continues to represent the established standard of care. However, not all melanoma cases are operable. This includes widespread lesions, lesions in anatomically or cosmetically sensitive areas (e.g., facial lesions where surgery might lead to significant disfigurement), and cases involving multiple suspicious moles that would require repeated surgical interventions. The adjuvant treatment options can be progressively individualized for such patients, considering the disease’s specific attributes and patient-related factors [8]. Adjuvant treatments are frequently employed to diminish the risk of recurrence, encompassing options such as chemotherapy, radiation therapy, hormone therapy, targeted therapy, or biological therapy [9].

The primary adjuvant treatment options typically are immune checkpoint inhibitors targeting cytotoxic T-lymphocyte antigen 4 (CTLA-4) or programmed cell death protein-1 (PD-1), along with small molecule BRAF (B-raf murine sarcoma viral oncogene homolog B1) inhibitors. However, the side effects of these therapies might be devastating; the effect is not durable, and hence, patients are more likely to relapse. In addition, these therapies are not always readily available, possibly delaying their initiation for months [10]. Furthermore, there are currently no established topical medications to complement systemic adjuvant therapies or to serve as alternatives in cases where systemic treatments are inaccessible, creating a need for research into novel topical treatments for melanoma.

The introduction of novel treatments for melanoma brings hope to reduce mortality in the upcoming decades. Likewise, developing topical therapy options is promising as most melanoma lesions are detected on the skin [11]. Various topical formulations have been attempted to treat melanoma, including 5-fluorouracil (5-FU) [12], piplartine [13], curcumin [14], and dacarbazine [15]. However, while these drugs can potentially treat melanoma, they are in experimental stages, and there is limited information about their performance in both in vivo studies and clinical settings. Moreover, given the complexity of determining antitumor efficacy through in vivo studies, topical skin treatments for melanoma are not widely accepted for treating patients [16]. The development of new strategies to treat melanoma can significantly contribute to the progression of the disease, increase the response to first-line treatments, and diminish treatment relapse.

Cantharidin (3a,7a-Dimethylhexahydro-4,7-epoxyisobenzofuran-1,3-dione; formula: C_10_H_12_O_4_; CAS-No.: 56-25-7), a defensive agent produced by beetles of the Meloidae family for the protection of fertilized eggs, is a terpenoid compound widely utilized in traditional Eastern medicine. Western dermatology has employed it as a topical treatment for furuncles, piles, warts, and molluscum contagiosum. While it is relatively safe when used as a colloidal solution in concentrations of 0.7–0.9%, its application is restricted to the dermatologist’s office or performed under direct supervision [17]. The application of cantharidin to the skin causes the release of neutral serine proteases, which trigger desmosome plaque degradation and subsequently result in the detachment of tonofilaments from desmosomes. This leads to intraepidermal blistering, acantholysis, and non-specific lysis of the epidermis. Dermatologists monitor and control the area and intensity of blistering by washing the affected area with soap and water, ensuring that blisters heal within 4 to 7 days without scarring.

Norcantharidin (NCTD; 7-oxabicycloheptane-2,3-dicarboxylic acid; formula: C_8_H_8_O_4_; CAS-No.: 29745-04-8) is a demethylated analog of cantharidin, that has been proposed as a chemotherapeutic agent with allegedly reduced toxicity. Both cantharidin and norcantharidin have been shown to inhibit protein phosphatases PP1 and PP2, which are crucial for regulating the cell cycle [18]. Additionally, the anhydride moiety of norcantharidin has been proposed as the active site responsible for its antitumor activity, which has been tested in various types of cancer, including neuroblastoma and glioblastoma [19]. In vitro evidence increasingly supports the effectiveness of NCTD as an oncological medication in several types of cell lines, such as colorectal, epithelial, ovarian [20], hepatocellular [21], esophageal, gastric, lung [19], and non-Hodgkin lymphoma [17]. Few attempts have been made to develop a clinical application of NCTD [22]. However, the most relevant was described by the Lixin study, which developed NCTD-containing microspheres for intradermal injection [23]. In 2020, a phase I clinical trial was submitted to evaluate NCTD-containing microspheres for injection in patients with solid tumors. However, the specific type of tumor was not disclosed, and the results have not been posted to this date [24].

In cutaneous melanoma, proliferating melanocytes are predominantly located in epidermis. Drug transport through the skin primarily occurs via two pathways: transcellular absorption, which involves movement through keratin-packed corneocytes by partitioning in and out of cell membranes, and intercellular absorption, which takes place around the corneocytes in the lipid-rich extracellular regions. Topical nanoemulsions are colloidal biphasic liquid-in-liquid systems composed of a hydrophilic phase and a hydrophobic phase, where one liquid phase is dispersed as nanosized droplets (<500 nm) in the other and stabilized by surfactants. They can be classified as either oil-in-water (O/W) or water-in-oil (W/O) [25]. In a previous study, our team developed and characterized a norcantharidin-loaded nanoemulsion with optimum droplet size, stability, and in vitro drug release profile. Furthermore, the antiproliferative effect against B16F1 melanoma cells was demonstrated in vitro [26]. A suitable formulation for delivering the drug to the epidermis should include features that ensure the active ingredients effectively reach the target area. Additionally, it should enable the localized application of larger drug quantities while minimizing systemic effects, allowing for prolonged treatment periods and ultimately improving therapeutic outcomes.

The inclusion of pentoxifylline (PTX) in this study stems from our research group’s prior investigations into its therapeutic potential for melanoma treatment. PTX is a methylxanthine derivative with well-documented anti-inflammatory and vasodilatory properties, which have been shown to modulate the tumor microenvironment. These characteristics make PTX a promising candidate for enhancing the therapeutic efficacy of existing treatment strategies. By combining PTX with the NCTD-containing nanoemulsion, we aimed to explore potential synergies between the two compounds. Although PTX has not been used for melanoma treatment, it could complement the cytotoxic effects of NCTD by improving vascular flow or immune cell infiltration. Moreover, microneedling in this study was strategically included as a physical enhancement method to improve skin penetration of the nanoemulsion. By creating microchannels in the outermost layer of the skin, microneedling facilitates deeper delivery of active compounds into the tumor microenvironment, allowing us to evaluate the penetration ability of the nanoemulsion further. Given the heterogeneous nature of melanoma, in vivo studies are necessary to determine early tumor onset progression, expansion, invasion, inflammatory and immune response, and metastasis propagation, complex features to evaluate in vitro [27].

## 2. Results

### 2.1. Nanoemulsion Formulation

The nanoemulsion was formulated according to the Hydrophilic-Lipophilic Balance (HLB) requirements, ensuring proper proportions of ingredients in the oil phase. The typical “cold cream” O/W emulsion ratio was maintained to achieve the necessary hydrophilic-lipophilic balance. The oil phase included almond oil (0.7 g) as an emollient, cetyl alcohol (0.2 g) serving as both an emulsifier and thickening agent, ceteareth-12 wax (0.2 g) as a hydrophobic surfactant, stearic acid (0.3 g) for texture, and glyceryl monostearate (0.2 g) as a lubricant. The water phase was composed of glycerin (0.6 g) as a humectant, urea (0.5 g) as a hydrating agent, polysorbate 80 (0.05 g) as a hydrophilic surfactant, Carbopol^®^ 940 (0.01 g) as a rheological modifier, and milliQ water to complete the mixture. Preparing the developed nanoemulsion followed standard procedures commonly used in dermatological product development. The active ingredient was added only after the temperature dropped, preventing degradation. The formulated nanoemulsion was a non-Newtonian white fluid characterized by a droplet mean size of 117 nm ± 1.2 S.D, a polydispersity index of 0.26, and a pH of 6.5, consistent with previous results.

### 2.2. Tumor Induction and Application of Treatments

The initial tumor inoculations proceeded without significant issues regarding inoculation. Within the first 3–4 days post-inoculation, the tumors began to visibly develop, with individual lesions emerging and growing at similar rates. Upon application of the treatments, the nanoemulsion demonstrated a favorable consistency, forming a thin, uniform layer over the affected area that was rapidly absorbed. When combined with microneedles, which act as physical enhancers by creating micro-channels in the outermost skin layer, occasional deep perforations were observed, leading to bleeding and subsequent callous formation in some cases. Despite these challenges, the nanoemulsion showed promising potential for effective delivery, highlighting the need for further optimization of the microneedle application technique to minimize adverse effects and enhance therapeutic outcomes.

### 2.3. Tumor Progression

In the initial photographic session capturing induced melanoma lesions at the twenty-one-day mark of the treatment, discernible disparities in tumor progression were observed. The control group showed a more pronounced advancement of the tumor lesion, with higher pigmentation. In the NCTDNem treatment group, there were varying degrees of tumor progression, ranging from advanced to those with imperceptible pigmentation. The NCTDNem + MD group displayed an overall reduction in tumor progression among individuals. Similarly, diminished progression was observed in the pentoxifylline-supplemented group (NCTDNem + MD + PTX) (Figure 1).

By the thirtieth day of treatment, the tumor progression observed at the 20-day mark continued. The control group exhibited increased tumor growth between the two photographic sessions, with notable peripheral redness and swelling. In the NCTDNem group, tumors varied widely, spanning from advanced progress to lesions with controlled tumor size. The NCTDNem + MD group showed minimal tumor growth; in some instances, significant macroscopical depigmentation of the tumor was observed. However, in the NCTDNem + MD + PTX group, there was a reversal of the previously observed trend (Figure 2).

The trend of tumor growth among the different treatment groups was reflected in the number of surgical excisions conducted in each group. The control group required the most surgical excisions (*n* = 8), followed by the NCTDNem group (*n* = 4), the NCTDNem + MD + PTX group (*n* = 3), and the NCTDNem + MD group (*n* = 2). The surgical excision percentages show a clear reduction in surgical interventions required across the treatment groups compared to the control group. These differences were statistically significant (*p* < 0.01), indicating a clear impact of the treatments on tumor growth. While the mean weight values of the groups did not show any statistical significance, there was a noticeable weight gain in the NCTD Nem + MD and the NCTD Nem + MD + PTX group, in contrast to the NCTDNem and control groups, which showed weight loss during the assay. Additionally, there were no significant differences in tumor volume among the groups (Figure 3).

### 2.4. Drug Content in Serum Samples

The drug content in serum samples varied across the different groups. The NCTDNem group had the lowest NCTD concentration, measuring 0.10 mg/mL. The NCTDNem + MD group had the second-highest NCTD concentration at 0.13 mg/mL. The NCTDNem + MD + PTX group had the highest NCTD concentration, measuring 0.18 mg/mL. As expected, no traces of either NCTD or PTX were detected in the control group. The group that was administered PTX exclusively had a PTX concentration of 1.23 × 10^−3^ mg/mL in serum (Figure 4).

### 2.5. Histopathological Analysis

Histopathological analysis revealed that all procured tumors exhibited typical characteristics of malignant epithelioid melanoma. Tumor size within the field of view was consistent across all groups. While the number of mitoses per high-power field did not show a statistically significant difference between the groups, the lowest mitotic count was observed in the NCTDNem + MD group. Additionally, this group displayed increased lymphocyte infiltration and signs of possible tumor regression. A complete summary of the optical field-of-view analysis is presented in Figure 5.

Particle analysis highlighted areas with melanin deposits and cellular nuclei in micrographs. According to the results, both groups had practically similar particle counts, indicating comparable cellularity. Furthermore, although no statistical significance was found regarding particle average size, there is a tendency for larger particles in the control group, which could imply more extensive melanin deposits. Ultimately, this aligns with the finding of a higher total particle area in the control group, indicating increased melanin production and a more significant pigment presence (Figure 6).

## 3. Discussion

The current World Health Organization classification of skin melanoma is based on morphological aspects and archetypical patterns of clinical and histological nature. Following a biopsy of a suspected melanoma lesion, confirmation of the diagnosis is necessary, which includes assessing the lesion’s size and the number of mitoses in 1.0 mm^2^ field of view. This evaluation helps determine the prognosis based on the lesion’s invasive capacity and treatment, whether surgical excision of adjuvant therapy [28]. Additionally, advancements in fast-growing sequencing technologies are identifying recurrent mutations in specific oncogenes. Many of these mutations are significantly associated with specific clinical or histopathological subsets of lesions, strongly suggesting biologically distinct types of melanocytic neoplasms. Consequently, while intraepithelial features do not necessarily indicate that the cell of origin is within the epithelium, the mutational burden observed in non-glabrous skin melanoma suggests that many originate from epidermal melanocytes [2]. Properly understanding skin melanoma tumor growth requires the development of models that integrate as many factors as possible. These models should emulate the current landscape of diagnosis and treatment to help create new and effective treatment strategies [27].

Topical pharmaceutical formulations aim to deliver more drug to targeted skin locations while minimizing systemic uptake. This approach offers several advantages, including minimal invasiveness, easy application, consistent pharmacokinetics, and improved local bioavailability by eluding first-pass metabolism. While the viable epidermis can retain topically administered drugs, limiting their dissemination beyond the epidermal layer, the stratum corneum is the bioavailability-controlling membrane for transporting xenobiotics across the skin. Therefore, local cutaneous bioavailability may not reflect systemic availability. Moreover, topical doses are often small, and the required concentration is often unknown [29]. Many melanoma-implanted murine models have been developed to study topical treatments with physical skin penetration strategies. Among these models are solid microneedle patches, liposomes, nanoemulsions, and transferosomes, to name a few [16]. Given the use of a nanoemulsion and the employability of the drug, a study that caught our attention assessed the in vivo effectiveness of a dacarbazine-loaded nanoemulsion that reduced melanoma growth [15]. Furthermore, an imiquimod-oleic acid prodrug cream induced toxicity and apoptosis in B16F1 synergenic C57BL murine model [30].

Norcantharidin is the demethylated analog of cantharidin. It is an experimental drug, and its use in humans has not been well established, as it has not been approved by any regulatory entity. In a preliminary toxicological study in BDF1 mice, we established a lethal dose 50 (LD_50_) of 8.8 mg/Kg and a therapeutic safe dose of 3 mg/Kg [31]. Notably, the concentration of NCTD found in serum samples was significantly lower than the safe dose proposed, suggesting that our formulation could support longer therapeutic schemes without exceeding toxicity thresholds. This is crucial as it indicates that while the tumor is exposed to a higher localized drug concentration, systemic exposure remains minimal, potentially reducing systemic toxicity risk. Furthermore, in contrast to the adverse effects reported, where higher systemic doses of NCTD led to noticeable signs of toxicity, including liver and kidney stress and general weakness in the animals, our cohort displayed a much better tolerance to the treatment. Specifically, the mice in our study maintained their weight and showed weight gain, particularly in the pentoxifylline supplementation group. Nevertheless, given that the liver and kidneys were reported affected by long-term exposure to NCTD, even at low systemic levels, it could pose risks. Therefore, it would be cautious to include liver and kidney metabolic monitoring in future studies, especially when considering longer therapeutic schemes.

NCTD is primarily used in Eastern medicine, particularly in traditional Chinese practices. However, its administration in humans—whether orally, intravenously, or through local injection—significantly irritates the application site [32]. Currently, there is an ongoing clinical study investigating its use in treating solid tumors via intratumoral injection, but the results have not yet been disclosed [22]. Norcantharidin exerts its anti-tumor effects in melanoma through multiple molecular pathways. Primarily, NCTD acts as a potent inhibitor of protein phosphatases PP1 and PP2A, which are crucial for cell cycle regulation. Prior studies have demonstrated that NCTD inhibits tumor growth and induces apoptosis in various cell lines. In human osteosarcoma cells, NCTD showed dose-dependent inhibition of proliferation and induction of apoptosis through the c-Met/Akt/mTOR signaling pathway. This effect involved reducing the expression of the anti-apoptotic protein Bcl-2 and increasing the expression of the pro-apoptotic protein Bax [33]. Another study investigated NCTD’s impact on melanoma, finding that it suppresses tumor growth and inhibits metastasis by downregulating MMP-2 expression, a protein linked to cancer cell invasion. Additionally, NCTD treatment led to a decrease in the activity of NF-κB, a transcription factor associated with MMP-2 regulation [34]. Furthermore, in a previous study, we demonstrated that NCTD inhibits cell proliferation in B16F1 melanoma cells and reduces the metabolic rate of these cells [26].

An important aspect to consider is that NCTD has limited solubility in water, with a maximum of 2.5 mg/mL at pH 6 and 9.5 mg/mL at pH 9.5 [35]. Additionally, norcantharidin’s capacity to partition between hydrophilic and lipophilic viable tissues may restrict its effectiveness in drug delivery [16]. The most promising approach to address this issue involves an injectable emulsion containing NCTD-lipid nanospheres. However, the concentration achieved in this formulation was only 8 mg/mL [23]. In the current study, we used a nanoemulsion as the delivery system. Nanoemulsions are biphasic liquid systems consisting of hydrophilic and hydrophobic phases, where one phase is dispersed as nanosized droplets into the second liquid, stabilized by surfactants [36]. Therefore, they can be an effective pharmaceutical form for enhancing drug delivery [37]. This study also utilized the microneedle technique to enhance the penetration and localized delivery of the norcantharidin nanoemulsion to melanoma tumors. Optimization primarily involved selecting the appropriate microneedle length and application method to ensure consistent drug delivery while minimizing potential tissue damage. We initially attempted to use commercially available microneedle pens with a vertical stamping motion. However, we found these devices too harsh for the delicate tissue of the mice’s paws, resulting in deeper puncture wounds that were counterproductive to our goal of minimizing tissue damage. Consequently, we opted for a more controlled manual application technique, where the pressure applied during the microneedling process was carefully applied to avoid excessive bleeding, which was occasionally observed when the needles were applied too forcefully.

In a prior drug release study, our NCTD-containing nanoemulsion (3% *w*/*w*, 30 mg/mg final concentration) was shown to release 15 mg of NCTD per gram of emulsion over a span of 3 h [26]. In the present study, 0.1 g of the nanoemulsion were applied daily to the melanoma lesion, corresponding to approximately 3 mg of the formulation being administered to the affected area daily. Given the release profile observed in the prior study, this would result in the delivery of about 1.5 mg of NCTD to the lesion everyday over the course of 30 days. The results showed that this dosage effectively reduced tumor size or delayed growth. Moreover, on average, the amount of NCTD found in the serum samples of treated groups was 0.15 mg per mL, indicating that approximately one-tenth of the applied dose was detected in serum. Also, applying the nanoemulsion with a microneedle pen would likely increase drug penetration and augment its effectiveness. The efficacy of the microneedle application was confirmed through the observed reduction in tumor size and the increased bioavailability of the drug in serum samples, compared to groups where the nanoemulsion was applied without microneedles. However, the variability in the extent of drug penetration suggests that further refinement of the microneedle application technique may be beneficial.

The histopathological analysis revealed that the selected syngeneic graft model is suitable for studying melanoma, as all tumors exhibited epithelial characteristics and were correctly located. Moreover, the homogeneity of tumor size across samples also indicates the model’s fitness. However, given the syngeneic nature of the lesion, we cannot fully assess invasiveness, as this also depends on the intrinsic properties of the cell line used. In addition to size, other histopathological characteristics that can predict the response to current melanoma treatments include the mitotic rate and lymphocytic infiltration within the lesion. Although the mitotic rate may also be an inherent characteristic of the cell line, the lowest rate was observed in the NCTD Nem + MD group, which received the highest dose of NCTD. Interestingly, the highest lymphocytic infiltration was observed in the same group, suggesting that NCTD may function as an immunomodulator. These findings indicate that immune system cells likely contribute to tumor reduction in conjunction with the drug’s effects, highlighting the potential of NCTD as an adjuvant treatment. A previous study from our research group demonstrated that combined treatments of PTX and NCTD administered intragastrically (60 mg/kg + 0.75 mg/kg) in DBA/2 mice significantly recruited lymphocytes and iNKT cells to the tumor site [38]. However, the PTX-complemented group demonstrated an unexpected reversal in tumor growth reduction in our study. This may be due to a pharmacological interaction and/or the increased blood filtration and clearance associated with PTX’s hemorheological effects.

Similarly, the reduced pigment in the treated groups may indicate diminished melanin production or a reduced tumoral mass. Additionally, probable regression was observed in the NCTD nanoemulsion group treated with a microneedle pen. Regression in cancer is very rare; however, melanoma, neuroblastoma, and lymphoma may show remission more frequently than any other cancer type, with a reported frequency of 1/100,000 [39]. Furthermore, a study on invasive breast cancer suggested that 22% of patients could experience tumor regression [40]. While further studies are needed, processing micrographs to evaluate melanin content could reveal insights into internal processes within the model [11]. Many studies have addressed melanoma syngeneic models and locoregional treatments [27,41]. Our study demonstrates that the norcantharidin (NCTD) nanoemulsion significantly reduces tumor growth in a syngeneic murine melanoma model. Although direct comparisons with established melanoma treatments such as immune checkpoint inhibitors (e.g., nivolumab, pembrolizumab) and small molecule inhibitors (e.g., vemurafenib, dabrafenib) are not viable due to the stage of our research and the pharmaceutical form employed, the results suggest that NCTD nanoemulsion could be a promising therapeutic option.

In cases where surgical excision is not feasible, the primary therapeutic alternatives for melanoma are systemic adjuvant therapies, including targeted and immunological treatments. These therapies, while critical, are often misunderstood due to the term ’adjuvant’ which may imply a complementary or secondary role. In reality, they serve as the cornerstone for treating inoperable melanoma. Despite their importance, there remains a critical gap in the availability of topical treatments that could complement systemic therapies or function as a standalone option, particularly for superficial melanomas. Additionally, the concept of ’co-adjuvant therapies’ has begun to emerge, presenting new opportunities to address unmet strategies in melanoma treatment. These therapies could aim to delay tumor growth, improve local tumor control, or enhance the efficacy of systemic adjuvant treatments by better-targeting tumor cells. By localizing the therapeutic effect, co-adjuvants may also help reduce the systemic side effects of traditional treatments and ultimately improve patient outcomes.

Topical treatments, such as imiquimod, have been explored for superficial melanomas, offering a non-invasive alternative with relatively good outcomes in specific cases. However, these treatments are often limited by the local laceration in long-term exposure. The nanoemulsion developed in our study is designed to enhance skin penetration as NCTD is a small molecule dissolved at a considerably high concentration, which may increase the exposure and local activity of the molecule while reducing systemic uptake. This could offer a more robust and consistent therapeutic effect over time. However, there is a lack of consensus on high-quality data, and varying levels of evidence. Moreover, fewer studies adhere to the management protocols used in clinical practice, resulting in inconsistent response rates and heterogeneous outcome measures [42]. This approach not only offers the potential for improved patient outcomes but also contributes to the growing body of evidence needed to support the use of alternative treatments in a broader clinical context [43].

In the present study, the management protocol aimed to adhere to clinical practice, incorporating dermatoscopic follow-up features, surgical incision (the primary treatment for melanoma stages I-III), and standard histopathological confirmation of the biopsy. Considering that various factors and patient characteristics may preclude surgery, it is important to assess potential locoregional therapies, as they may significantly affect progression-free survival. Moreover, these therapies should be evaluated not only for their effect on patient-rated outcomes but also for their feasibility and comparative health costs [8]. It is important to emphasize that our findings are based on preclinical models. While they provide valuable insights into the potential efficacy of NCTD nanoemulsion, further research is necessary to determine its safety, optimal dosage, and effectiveness in humans. Furthermore, exploring the combination of NCTD nanoemulsion with existing melanoma therapies could be an exciting avenue for future research. Given the ability of NCTD to inhibit protein phosphatases and its potential role in modulating the immune response, combining this treatment with immune checkpoint inhibitors or targeted therapies could lead to synergistic effects, enhancing the overall therapeutic outcome. Future studies should include clinical trials to evaluate the ORR, progression-free survival, and overall survival in patients treated with NCTD nanoemulsion.

## 4. Materials and Methods

### 4.1. Materials and Reagents

The materials and reagents used in this study included almond oil, Carbopol^®^ 940 (Lubrizol, Wickliffe, OH, USA), cetyl alcohol, glycerin, glyceryl monostearate, methylparaben, propylparaben, stearic acid, triethanolamine, polysorbate 80, urea (Merck KGaA, Darmstadt, Germany), and Eumulgin^®^ B1 (BASF SE, Ludwigshafen, Germany). The active compound, norcantharidin (NCTD), was obtained from Sigma-Aldrich (St. Louis, MO, USA). Cell culture reagents included DMEM, fetal bovine serum (FBS), glutamine, pyruvate, erythromycin, ampicillin, and trypsin-EDTA (Life Technologies, Carlsbad, CA, USA). For surgical procedures, isoflurane, ketamine, midazolam, stainless steel surgical forceps, and saline solution were used. Cyclophosphamide (internal standard), acetone, hydrochloric acid, and sodium citrate tubes were employed for serum drug extraction and analysis (Merck KGaA, Darmstadt, Germany). Histological analyses utilized hematoxylin, eosin, and paraffin embedding supplies.

### 4.2. Nanoemulsion Development

The norcantharidin nanoemulsion was developed using an oil-to-water weight ratio of 1:5, with the formulation designed based on the hydrophilic-lipophilic balance (HLB) requirements of the components. The oil phase included almond oil (0.79 mmol), cetyl alcohol (0.825 mmol), ceteareth-12 (0.28 mmol), stearic acid (1.05 mmol), and glyceryl monostearate (0.558 mmol). The water phase consisted of glycerin (6.52 mmol), urea (8.33 mmol), polysorbate 80 (0.038 mmol), triethanolamine (TEA, 1.34 mmol), and Milli-Q water to adjust the volume. The active ingredient, norcantharidin (NCTD), was incorporated at 1.65 mmol, achieving a final concentration of 3.0% *w*/*w* (30 mg/mL). The final emulsion was sonicated for 5 min with an ultrasonic processor at 5.0 watts to obtain the final nanoemulsion (Vibracell, Sonics & Materials, Newton, MA, USA). The nanoemulsion was further characterized by droplet mean size and polydispersity index (PDI) with a Zetasizer Nano Series ZS apparatus (Malvern Panalytical, Malvern, UK). Additionally, the pH was measured using a pH meter (AB315, Thermo Fisher Scientific, Waltham, MA, USA) (Figure 7).

### 4.3. Cell Culture

The immortalized B16F1 melanoma cells were obtained from the American Type Culture Collection. Cells were cultured using a mixture of DMEM culture medium, supplemented with 5% fetal bovine serum, 2 nM glutamine, 1% antibiotics (erythromycin at 10,000 U/mL and ampicillin at 10,000 U/mL), and pyruvate (all bought from Life Technologies, Carlsbad, CA, USA) in 75 cm^2^ flasks and placed under a humidified incubator at 37 °C with a CO_2_ concentration of 5% and an air saturation of 95% in the atmosphere. Passages were performed using a trypsin-EDTA solution (0.25% *w*/*v* −0.02%) in phosphate-buffered saline (PBS). In the ninth passage, cells were left to grow to over-confluence [44]. The culture medium was exchanged for a non-supplemented medium 24 h before the assay. After time elapsed, culture debris and remanent medium were discarded. Cells were detached, and 1 mL of supplemented medium was added to neutralize trypsin. Cell suspensions were counted using a 0.4% *w*/*v* trypan blue dye solution. After counting, cell solutions were transferred into a 10 mL polypropylene for centrifugation (1200× *g* for 5 min). Cell pellets were resuspended and adjusted to a concentration of 2 × 10^6^ cells per 1.0 mL with injectable water and loaded into 1.0 mL ultrafine syringes.

### 4.4. Syngeneic Graft Model

ICR mice (*n* = 32) were acquired from the National School of Biological Sciences animal facility. All mice of both sexes were evaluated at 4 to 6 weeks of age, with a weight greater than 20 g. Mice were maintained at 23 °C with a 12 h cycle of darkness and light. Additionally, a humidity level of 40–60% was maintained. The mice had ad libitum access to water and food [45]. Mice were held by the junction of the knee of the left hind leg, and 0.1 mL of the B16F1 cell solution was injected subcutaneously to obtain a final inoculation of 200,000 cells. The injection depth was carefully controlled using 1.0 mL ultrafine syringes (U-100 30G 1 mL/cc 5/16”, 8 mm) and inclining the needle at a 45° angle to ensure accurate placement within the subcutaneous tissue. Three days after the initial inoculation, mice were randomly assigned to four treatment groups [46]. Tumor volume was calculated using the following formula:$$Tumor\;volume=\frac{\pi }{6}\times Lenght\;\left(mm\right)\times {Width}^{2\;}(mm)$$

*Length* was taken as the tumor’s longest diameter, and *Width* is the shortest perpendicular diameter. Measurements were taken using digital calipers to ensure accuracy, and the volume was expressed in cubic millimeters (mm^3^).

### 4.5. Treatment Groups

The sample size was selected based on standard practices in preclinical studies for murine models, ensuring statistical power while adhering to ethical considerations of animal use. The treatments were administered once daily, starting on the third day after tumor inoculation and continuing for 30 days. The first group received as treatment the application of 0.1 g of the norcantharidin nanoemulsion (NCTDNem; *n* = 8) using a fine-tipped cotton swab where the tumor was inoculated. The second group was treated with the same amount of nanoemulsion, previously pressing the inoculation area with an adjustable Microneedling Pen with 36 nano pin cartridges (NCTDNem + MD; *n* = 8). The third group was orally treated with pentoxifylline (PTX) using a 60 mm mouse feeding needle at a dose of 60 mg/kg of body weight, in addition to the application of nanoemulsion with the microneedling pen (NCTDNem + MD + PTX; *n* = 8). All the groups were compared to the control group and no exclusions were made. The personnel administering treatments were aware of the group allocations to ensure proper delivery of interventions, while the researchers conducting tumor measurements, serum drug content analysis, and histopathological assessments were blinded to the group allocations throughout the study. On days 20 and 30 of the assay, melanoma lesions from all groups were photographed using a dermatoscope (DermLite, 3Gen Inc., San Juan Capistrano, CA, USA) for amplification and captured with a high-resolution 48 MP smartphone camera.

### 4.6. Surgical Intervention

On the 30th day of treatment, those individuals with melanoma tumors exceeding a diameter of 4 mm underwent surgical excision. For this purpose, mice were anesthetized in an induction chamber (RWD, vaporizer for isoflurane Mod. R5835, Sugar Land, TX, USA) saturated at 4.0% with isoflurane, complemented by 1.2 mg/kg of ketamine and 3 mg/kg of midazolam. A tourniquet was applied to the upper third of the leg below the femorotibial joint. Once circulation was reduced, the affected area was excised using stainless steel forceps and cauterized with a high-temperature electrocautery device (Bovie Clearwater, FL, USA). Post-operative care included cleaning the affected area and performing additional cauterization if bleeding was observed. Data is presented as the percentage of surgical excisions performed, mean weight (g), and tumor volume (mm^3^) per group.

### 4.7. Blood Samples

After surgical procedures, each group resumed their assigned treatments. The NCTD nanoemulsion was applied to the limbs of mice that underwent surgery. Conversely, the nanoemulsion was applied directly to the left hind leg, where the tumor was initially inoculated for those who did not undergo surgical excision. PTX administration followed the procedure outlined in the previous section. This procedure aimed to replicate post-operative care in a realistic clinical context or an extended treatment period exceeding 30 days and to determine drug serum content. After a ten-day interval, all mice were anesthetized according to the previously described conditions and then euthanized. Blood samples were obtained through cardiac puncture and collected in tubes containing sodium citrate (109 mmol/L) as an anticoagulant. Subsequently, samples were centrifuged at 1200× *g* for 5 min to separate the plasma.

### 4.8. Drug Content Determination

Both NCTD and PTX drugs were extracted from a 0.5 mL serum aliquot fortified with 25 µL of internal standard (Cyclophosphamide, 99.94% from MERK, Rahway, NJ, USA), 100 µL of 1 M hydrochloric acid, and 1.0 mL of acetone. The mixture was centrifuged at 10,000 rpm for 5 min, separating the supernatant from the precipitate. PTX was determined in the supernatant. Concurrently, the precipitate was utilized for NCTD determination after resuspension with 1.0 mL of acetone, followed by evaporation to dryness. The resulting residue was reconstituted with 150 µL of water. NCTD determination was determined by fluorescence spectroscopy with an excitation/emission wavelength of 300/450 nm, while PTX was determined by direct absorbance (280 nm) using a multi-mode microplate reader Synergy MX ((BioTek Instruments, Inc., Winooski, VT, USA) Biotek) in 96-well plates (Thermo Fisher Scientific, Waltham, MA, USA) [47].

### 4.9. Histological Analysis

Tumor samples underwent further processing using the paraffin inclusion technique and were stained with hematoxylin and eosin. Micrographs of 3-micron thick slices (Microtome RM2245, Leica Microsystems GmbH, Wetzlar, Germany) were captured at 20× magnification using optical microscopy (Binocular optical microscope Leica Microsystems). Micrographs were divided into two groups: the controls (*n* = 8) and the treated samples (*n* = 9), regardless of treatment. Micrographs were analyzed per field-of-view, and subsequently, size (cm), mitosis by high power field (HPF), lymphocyte infiltration, and regression (a phenomenon in which cancer lesions shrink naturally or using drugs) were assessed [48].

### 4.10. Micrographs Image Analysis

The multispectral micrographs were split into color channels: red, blue, and green. Channel blue was used to capture the visible light from 450–495 nm, and the threshold was adjusted to 28% using the max entropy filter. Afterward, a selection was created, and a particle analysis was performed concerning particle count, average size, and the total area measured by pixels. Additionally, a quantitative analysis at object and spatial levels was conducted using Fiji software (NIH, Bethesda, MD, USA, available online: https://imagej.nih.gov/ij/ (accessed on 1 February 2024) [49].

### 4.11. Statistical Analysis and Data Interpretation

Statistical analyses confirmed that this sample size was adequate to achieve a power of 80% (at an alpha level of 0.05) for detecting moderate to large effect sizes in key outcomes, including tumor size reduction and serum NCTD levels. All pertinent statistical analyses were conducted to ensure the robustness and validity of the results. Initially, normality tests, such as the Shapiro–Wilk test, were used to assess the data distribution’s symmetry, while linearity was examined for regression analysis. Fisher’s exact test was employed for categorical data. Specifically, the surgical excision counts among the treated groups due to its appropriateness for small sample sizes. Regression analysis with calibration curves was performed to estimate drug content in serum. The calibration curves included fixed amounts of NCTD (0.002 mg to 0.008 mg, Equation (1): y = −78.43 × 10^4^ + 11.27 × 10^3^; R^2^ = 0.993) and PTX (2 × 10^−5^ mg to 8 × 10^−5^ mg, Equation (2): y = 7150 + 0.039; R^2^ = 0.997). The limit of detection (LOD) and limit of quantification (LOQ) were then calculated based on the standard deviation of the response (σ) and the slope (S) of the calibration curves. For NCTD, the LOD was determined to be approximately 0.002 mg/mL, with an LOQ of 0.007 mg/mL. Similarly, for PTX, the LOD was calculated at approximately 0.021 mg/mL, with an LOQ of 0.063 mg/mL.

Different tests were applied based on the data distribution for continuous data obtained from histopathological and particle analyses. Size data is presented as the mean size of tumors in centimeters (cm) ± standard deviation (S.D). Mitoses are counted per 1 mm of high-power field (HPF). Lymphocyte infiltration (L.I) was evaluated qualitatively from less to more (<<). Regression (Reg) is presented as presence (+) or absence (−). The Student’s t-test was utilized for normally distributed data, whereas the Mann–Whitney U test was used for non-normally distributed data. This differentiation ensured the appropriate statistical approach was applied to each data set. Statistical comparisons were carried out using PAST software version 2.17c [50]. Plots were created with Sigma Plot (Version 11.0 Build 11.0.0.77, Systat Software, Inc., 2008, Chicago, IL, USA), and figures were generated using Publisher (Microsoft Corporation, Redmond, WA, USA). A *p*-value of less than 0.05 was considered statistically significant for all tests.

## 5. Conclusions

A novel NCTD-containing nanoemulsion that is systemically safe while allowing for higher doses to be applied directly to the tumor location and exerting the desired effect was successfully developed. This potential therapy improved therapeutic outcomes, such as tumor growth arrest observed during dermatoscopical follow-up, reduced surgical incision interventions, and positive histopathological features frequently used in clinical practice. This innovative scientific development, supported by robust evidence, could serve as an alternative in the current treatment of early-stage melanoma or as adjuvant therapy, improving disease progression outcomes and positively impacting the lives of melanoma patients. Our study aimed to address these needs by proposing a nanoemulsion-based therapy as a potential topical alternative or co-adjuvant. Such an approach could play a pivotal role in healthcare systems with limited access to systemic therapies, offering an affordable and readily deployable solution.

## Figures and Tables

**Figure 1 ijms-26-01215-f001:**
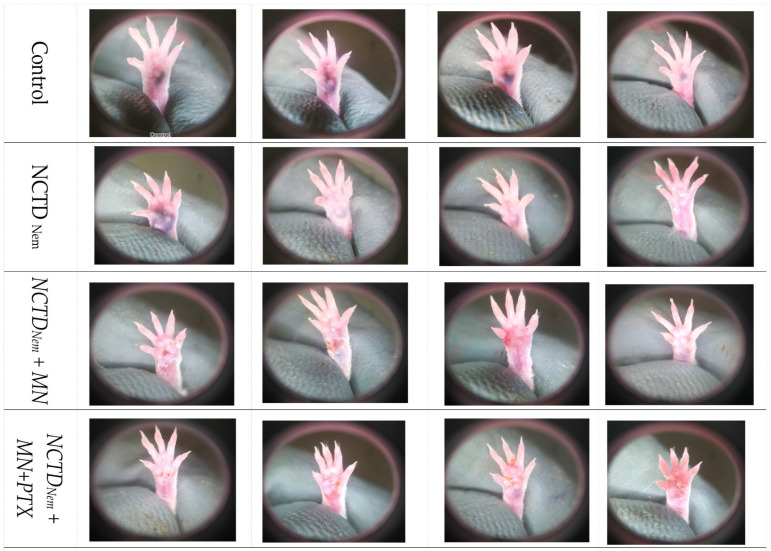
Representative Photographs of Melanoma Lesions After 20 Days of Treatment Across Different Groups. The images depict the condition of the melanoma lesions in four representative mice from each treatment group on the 20th day of treatment. Top row: Control group, which received no treatment, showing advanced tumor progression. Second row: NCTD nanoemulsion-treated group (NCTDNem), displaying varying degrees of tumor progression. Third row: NCTD nanoemulsion applied with microneedles (NCTDNem + MD), demonstrating a reduction in tumor size and progression. Bottom row: NCTD nanoemulsion with microneedle application supplemented with pentoxifylline (NCTDNem + MD + PTX), showing further reduction in tumor size and evidence of potential regression.

**Figure 2 ijms-26-01215-f002:**
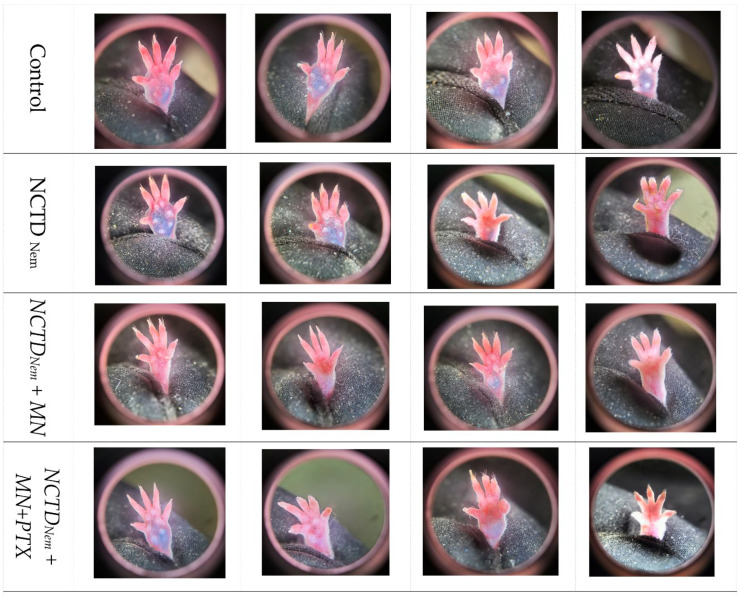
Representative Photographs of Melanoma Lesions After 30 Days of Treatment Across Different Groups. The images show the condition of the melanoma lesions in four representative mice from each treatment group on the 30th day of treatment. Top row: Control group, showing continued tumor progression with more prominent growth and pigmentation. Second row: NCTD nanoemulsion-treated group (NCTDNem), where varying levels of tumor progression are observed, including some cases of advanced growth. Third row: NCTD nanoemulsion applied with microneedles (NCTDNem + MD), displaying minimal tumor growth and signs of potential tumor regression. Bottom row: NCTD nanoemulsion with microneedle application supplemented with pentoxifylline (NCTDNem + MD + PTX), showing further reduction in tumor size and less pigmentation compared to other groups.

**Figure 3 ijms-26-01215-f003:**
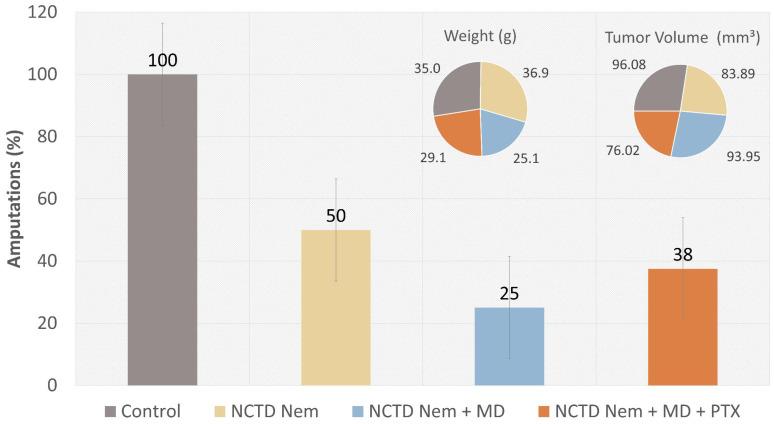
Impact of Treatments on Surgical Excision Rates, Group Mean Weight, and Macroscopic Tumor Volume. The vertical bars represent the percentage of surgical excisions within each treatment group, with standard error indicated. The pie charts illustrate the mean weight (left pie chart) and macroscopic tumor volume (right pie chart) across treated groups. The groups include Control (no treatment), NCTD nanoemulsion (NCTDNem), NCTD nanoemulsion applied with microneedle (NCTDNem + MD), and NCTD nanoemulsion applied with microneedle and supplemented with pentoxifylline (NCTDNem + MD + PTX).

**Figure 4 ijms-26-01215-f004:**
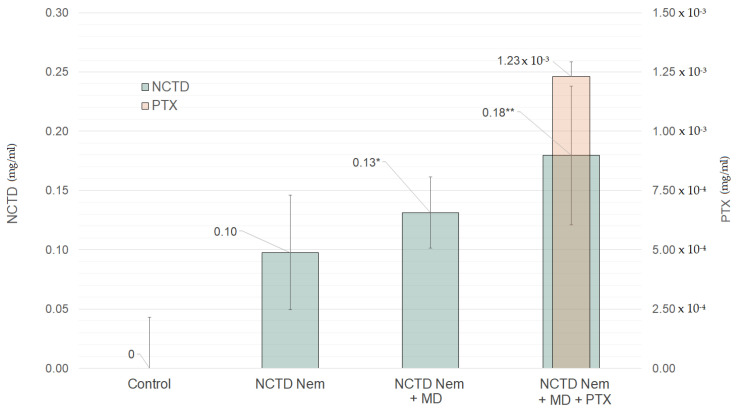
Systemic bioavailability of NCTD and PTX in serum samples. The data is presented as milligrams (mg) per mL of serum across different treatment groups. The green bars represent norcantharidin (NCTD) concentration, while the orange bar represent the groups’ pentoxifylline (PTX) concentration. The groups are Control, Norcantharidin nanoemulsion (NCTDNem), Norcantharidin nanoemulsion applied with microneedle (NCTDNem + MD), and Norcantharidin nanoemulsion applied with microneedle supplemented with pentoxifylline (NCTDNem + MD + PTX). Statistical significance: *: *p* < 0.05; **: *p* < 0.01.

**Figure 5 ijms-26-01215-f005:**
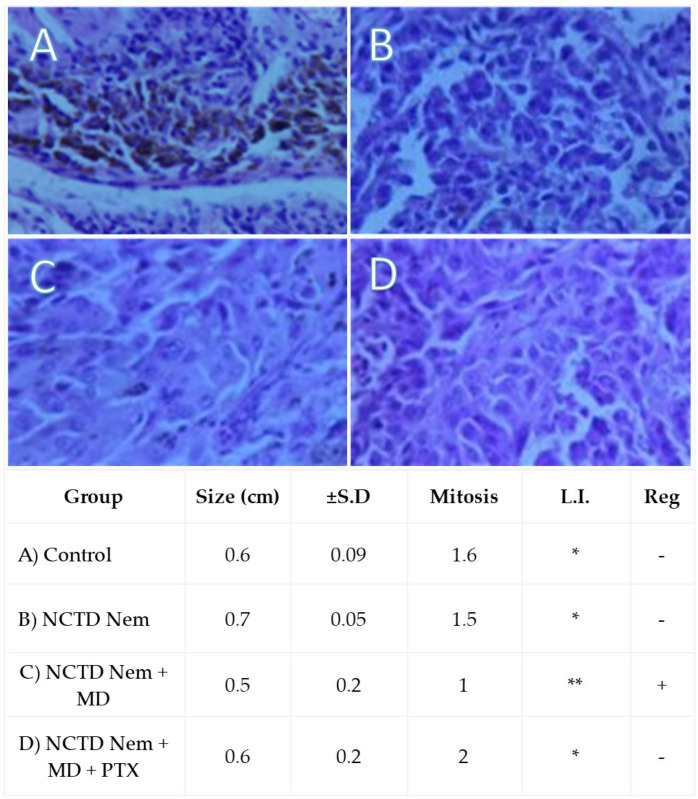
Representative high-resolution micrographs of H&E-stained melanoma samples. Size data is presented as the mean size of tumors in centimeters (cm) ± standard deviation (S.D.). Mitoses are counted per 1 mm of high-power field (HPF). Lymphocyte infiltration (L.I.) is evaluated qualitatively, with the scale ranging from less to more (* < **). Tumor regression (Reg) is indicated as presence (+) or absence (−). Groups are (**A**) Control, (**B**) Norcantharidin nanoemulsion (NCTDNem), (**C**) Norcantharidin nanoemulsion applied with microneedle (NCTDNem + MD), and (**D**) Norcantharidin nanoemulsion applied with microneedle and supplemented with pentoxifylline (NCTDNem + MD + PTX).

**Figure 6 ijms-26-01215-f006:**
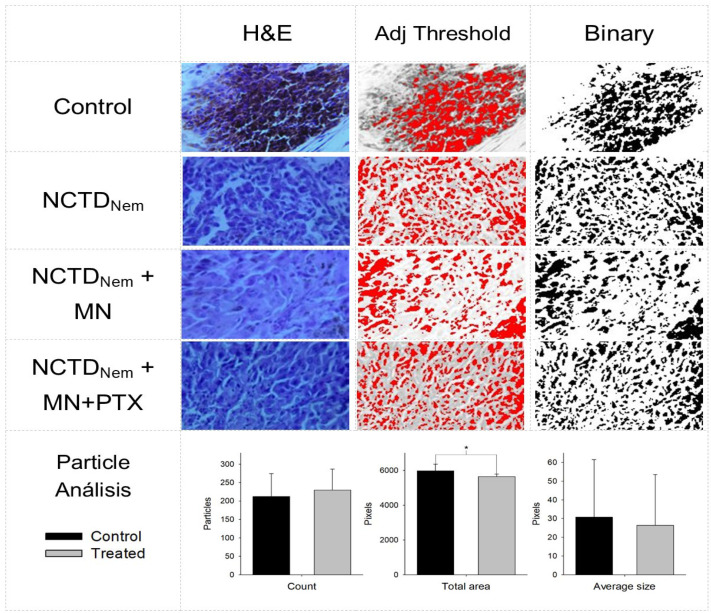
Multispectral analysis of micrographs from tumor lesion fixed in and tainted histologically. First column hematoxylin and eosin (H&E) tainted samples. Second column: adjusted threshold using the image blue channel. Third column: Binary selection of particle analysis. Groups are Control, Norcantharidin nanoemulsion NCTDNem, Norcantharidin nanoemulsion applied with microneedle (NCTDNem + MD), Norcantharidin nanoemulsion applied with microneedle and supplemented with pentoxifylline (NCTDNem + MD + PTX). Bottom: Particle analysis presented as particle count, total area (pixels), and pixel average particle size (pixels). Groups are Control v. Treated. Significance *: *p* < 0.05.

**Figure 7 ijms-26-01215-f007:**
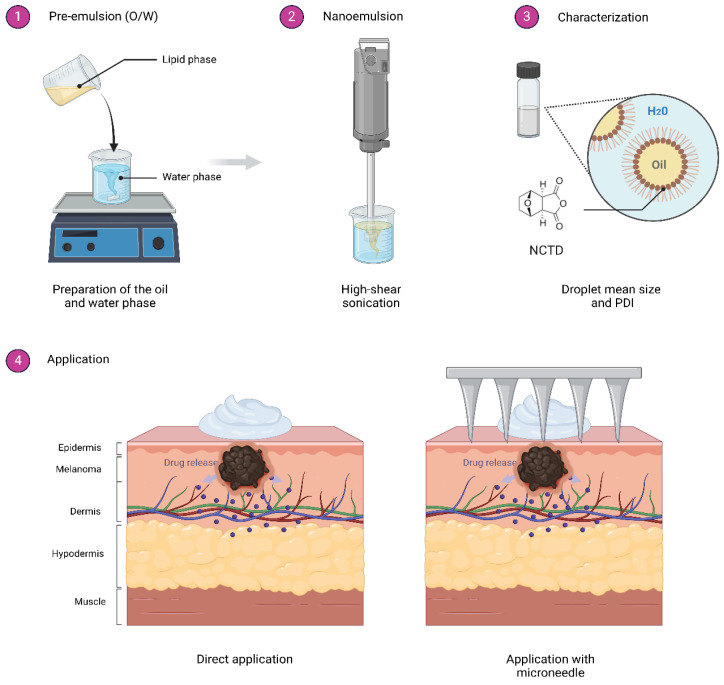
Nanoemulsion preparation, characterization and application to the syngeneic graft inoculation area. (**1**) Pre-emulsion preparation by mixing the oil phase into the water phase. (**2**) Nanoemulsion obtention through high-energy sonication. (**3**) Characterization of the nanoemulsion in droplet size and PDI. (**4**) Different forms of application forms, direct and using microneedle.

## Data Availability

Data and full photographic records supporting reported results are available under request.

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
