# Peer review of "Evaluation of a Norcantharidin Nanoemulsion Efficacy for Treating B16F1-Induced Melanoma in a Syngeneic Murine Model"

_ijms, 2025, doi:10.3390/ijms26031215_

Round 1
Reviewer 1 Report
Comments and Suggestions for Authors
In the manuscript “Evaluation of a Norcantharidin Nanoemulsion Efficacy for Treating B16F1-Induced Melanoma in a Syngeneic Murine model”,the authors aimed to develop a new treatment and a complementary option.While actually the main treantment style of melanoma is surgical excision and there is no other alternative medical treatment,authors think that not all melanoma cases are candidates for surgical procedures,while authors didn't expain clearly that surgical procedures is not needed under what circumstances.There have another questions except that.
1.In the experiments of B16F1-Induced Melanoma in a Syngeneic Murine model, there are only four groups, control group, NCTD nanoemulsion-treated group,NCTD nanoemulsion applied with microneedles and NCTD nanoemulsion with microneedles application supplemented with pentoxifyline. I don't know the meaning of the forth group, the aime of pentoxifyline with NCTD and microneedles.
2. Pentoxifyline is a positive medicine of melanoma, Is it a medication that has already been used clinically?
3.Photographs of melanoma lesions after differents days of treatment across the four groups only presented four samples, we can consider it is not enough only with representative photographs.
4.Figure 4 did not express clealy and the figure is not good.
5.The authors determined the drug content in serum samples, including DCTD and PTX, Why do the authors determined the PTX content in serum samples?
6.The research of this manuscription is a little easy.
Reviewer 2 Report
Comments and Suggestions for Authors
This article evaluated the efficacy of a norcantharidin nanoemulsion in treating B16F1-induced melanoma in a syngeneic murine model. The study is highly novel, well-written, clear, and impactful except for a few minor issues.
1. line 102: "Few attempts have been 102 made to develop a clinical application of NCTD have been made." seems need to delete the repeat "have been made" .
2. Regarding the model construction, where exactly are the tumor cells injected? Is there a specific depth requirement for the injection?
3. In the materials and methods section, the frequency of drug intervention is not clearly specified. I only learned in the discussion section that the drug was administered daily from the third day after inoculation until day 30. Please clarify this in the materials and methods section.
4.Given that the actual administration period is 27 days, is there an error in the drug dosage calculation in the discussion? Or does your mention of 30 days refer to continuous drug administration for 30 days, rather than 30 days after tumor inoculation? Please clarify.
5. About histopathological analysis, could you clarify how the differences in lymphocyte infiltration between groups were determined? Are there high-resolution images available to support this?
Reviewer 3 Report
Comments and Suggestions for Authors
The manuscript authored by Martínez-Razo et al. describes the preparation and in vivo evaluation of a nanoemulsion containing norcantharidin (NCTD) for the treatment of melanoma. This manuscript seems to be the continuation of an article recently published by the same authors (Pharmaceuticals 2023, 16, 4, 501), in which the same nanosystem was evaluated in vitro. The manuscript is well written, and the data, which is convincing, is also well discussed. The topic fits within the journal’s scope. There are some comments/suggestions that the authors should address during the revision process.
Minor comments
- Page 3, line 102: “Few attempts have been made to develop a clinical application of NCTD have been made.” Please correct this sentence.
- Section 4. Materials and methods: Please describe the materials used in this study.
- Nanoemulsion development: Please specify the volume of the oil and water phases, including the molar ratio of all components used in the formulation.
- Please include the formula used for the measurement of tumor volume in the materials and methods section.
Major comments
- Although the authors carried out part of the characterization of the nanoemulsion, they should include a TEM micrograph and a DLS size distribution to confirm the morphology and quality of the formulation.
- Page 4, Line 143: Please include the SD of the droplet size.
- Figure 1: The authors should study the effect of NCTD alone or in combination with pentoxifylline as control experiments. This would confirm whether the use of nanoemulsions may potentiate the therapeutic effect and reduce melanoma lesions when compared to the drug alone.

Round 2
Reviewer 3 Report
Comments and Suggestions for Authors
The authors have addressed the reviewer's comments/suggestions and therefore I recommend this manusript for publication in IJMS